# In Vivo Tracking and 3D Mapping of Cell Death in Regeneration and Cancer Using Trypan Blue

**DOI:** 10.3390/cells13161379

**Published:** 2024-08-20

**Authors:** Nicole Procel, Karen Camacho, Elisabeth Verboven, Isabel Baroja, Priscila A. Guerrero, Hanne Hillen, Carlos Estrella-García, Nicole Vizcaíno-Rodríguez, Leticia Sansores-Garcia, Ana Santamaría-Naranjo, Andrés Romero-Carvajal, Andrés Caicedo, Georg Halder, Iván M. Moya

**Affiliations:** 1Cancer Research Group, Faculty of Engineering and Applied Sciences, Universidad de Las Américas, Quito 170124, Ecuador; 2Department of Cell and Molecular Biology, Karolinska Institutet, 17165 Stockholm, Sweden; 3VIB Center for Cancer Biology and KU Leuven Department of Oncology, KU Leuven, 3000 Leuven, Belgium; 4Faculty of Health Sciences and Medicine, Universidad de Extremadura, 06800 Mérida, Spain; 5Laboratorios Multidisciplinarios de Ciencias Biológicas y Químicas, Universidad de Las Américas, Quito 170513, Ecuador; 6Escuela de Ciencias Biológicas, Pontificia Universidad Católica del Ecuador, Quito 170525, Ecuador; 7Colegio de Ciencias de la Salud, Escuela de Medicina, Universidad San Francisco de Quito USFQ, Quito 170901, Ecuador

**Keywords:** in vivo cell death tracking, Trypan Blue labeling, organ regeneration, anticancer therapy assessment, ischemia-reperfusion injury, cholangiocarcinoma, necrosis, apoptosis

## Abstract

Tracking cell death in vivo can enable a better understanding of the biological mechanisms underlying tissue homeostasis and disease. Unfortunately, existing cell death labeling methods lack compatibility with in vivo applications or suffer from low sensitivity, poor tissue penetration, and limited temporal resolution. Here, we fluorescently labeled dead cells in vivo with Trypan Blue (TBlue) to detect single scattered dead cells or to generate whole-mount three-dimensional maps of large areas of necrotic tissue during organ regeneration. TBlue effectively marked different types of cell death, including necrosis induced by CCl_4_ intoxication in the liver, necrosis caused by ischemia-reperfusion in the skin, and apoptosis triggered by *BAX* overexpression in hepatocytes. Moreover, due to its short circulating lifespan in blood, TBlue labeling allowed in vivo “pulse and chase” tracking of two temporally spaced populations of dying hepatocytes in regenerating mouse livers. Additionally, upon treatment with cisplatin, TBlue labeled dead cancer cells in livers with cholangiocarcinoma and dead thymocytes due to chemotherapy-induced toxicity, showcasing its utility in assessing anticancer therapies in preclinical models. Thus, TBlue is a sensitive and selective cell death marker for in vivo applications, facilitating the understanding of the fundamental role of cell death in normal biological processes and its implications in disease.

## 1. Introduction

Trypan Blue (TBlue) is a dye that selectively penetrates the cell membrane of dead cells and is commonly used to evaluate cell viability in vitro [1]. When exposed to cytotoxins or other stresses, cells may undergo various types of cell death, such as apoptosis or necrosis, rendering them permeable and capable of incorporating TBlue. In dead cells, TBlue binds to intracellular proteins, resulting in a distinctive blue coloration, while viable cells remain uncolored due to membrane impermeability [2]. Additionally, TBlue exhibits fluorescence within the spectrum of 630–720 nm when stimulated at 580–640 nm, enabling the fluorescent labeling of dead cells [3]. This property allows the combined labeling of dead cells by TBlue and different populations of living cells by specific fluorescent markers. These features collectively make Trypan Blue an ideal tool for discerning between living and dead cells.

Despite its ease of use and cost-effectiveness, the use of TBlue has been predominantly restricted to in vitro studies involving dissociated cultured cells, ex vivo analysis of biopsied samples, or as a visual aid to mark dead tissue during surgery [4,5,6,7]. In contrast, here we describe the use of TBlue to capture ongoing cell turnover processes in a mouse in vivo and to reveal the fate of dead cells in terms of perdurance, rates of cell death, and dead cell clearance. We utilized TBlue to label and track dead cells in living mice with injured or cancerous tissues, demonstrating the sensitivity and selectivity of TBlue as an in vivo marker for cell death. We used the liver as a model organ to investigate the ability of TBlue to label cell death in living animals because cell death plays an integral role in organ regeneration and carcinogenesis. We used the liver extensively because it offers several advantages for the study of cell death. First, one can use hepatotoxins to induce predictable and stereotypical patterns of hepatocyte death. Second, the perdurance of dead cells is transient and tightly regulated during liver regeneration, which allows testing the sensitivity of TBlue labeling in a dynamic process. Third, several aspects of the biology of chronic liver disease and cancer are characterized by an increase in cell death. However, the utility of this method is not restricted only to the liver, as TBlue labels dead cells induced by ischemia-reperfusion in the skin, organ atrophy caused by fasting, and chemotherapy-induced toxicity in the thymus. Thus, TBlue allowed us to mark and trace dead cells in different organs and to estimate the dynamics of cell death and clearance in different settings of regeneration and disease.

## 2. Materials and Methods

### 2.1. Mouse Strains and Allele Deletion

*Osteopontin-iCreER*^T2^*;Yap*^flox/flox^*;Taz*^flox/flox^ mice were generated by crossing *Yap*^flox/flox^*;Taz*^flox/flox^ (generated by Erik Olson) to *Opn-iCreER*^T2^ mice. BALB/c and C57BL/6 mice were from Charles River Laboratory (Wilmington, MA, USA). Controls matched for sex and age were littermates. Tamoxifen (#13258. Sanbio, Uden, The Netherlands) was intraperitoneally injected on 5 consecutive days (1.6 mg/kg in oil), followed by a 3-week washout period. Mouse experiments, housing, and feeding conditions were approved by the Institutional Ethical Commission for Animal Research at Katholieke Universiteit Leuven (Belgium) and at the University of San Francisco Quito (Ecuador).

### 2.2. Carbon Tetrachloride (CCl_4_) Administration

Mice received a single intraperitoneal injection of carbon tetrachloride (CCl_4_; 1 mL/kg, #319961 Sigma-Aldrich, Merck KGaA, Darmstadt, Germany) diluted in corn oil (1:8);. At 12 to 96hoursafter CCl_4_ administration, mice were euthanized with carbon dioxide, and livers were collected.

### 2.3. Ischemia-Reperfusion Injury

We used magnets to simulate ischemia and reperfusion in the adult mouse ear, as described by Goh et al. (2016) [8]. In brief, after 90 minutes of ischemia, the clamping magnets were removed to allow reperfusion. Then, the volume required for a dose of 25 mg/kg of 0.4% Trypan Blue was administered intraperitoneally, and mice were sacrificed 2 hours after Trypan Blue administration. Ear tissues were fixed in 4% paraformaldehyde (PFA) for 24 hours, sectioned at 70 µm with a Leica-VT 1000S Vibratome (Leica Biosystems, Nussloch, Germany), and further processed for immunofluorescent analysis. 

### 2.4. Thymus Atrophy and Toxic Injury

To induce thymus cell death and atrophy, male BALB/c mice of about 30 g of weight were fed normal chow or fasted for a period of 48 hours [1]. Given that mice are coprophagic, during the period of fasting, all mice were housed in individual cages with clean bedding. Water was provided ad libitum*,* and all mice were closely monitored for signs of discomfort. Upon the culmination of fasting and 2 hours prior to sacrifice, mice were injected intraperitoneally with 200 µL of Trypan Blue. To create thymocyte cell death in response to chemotherapy, male BALB/c mice of the same weight and age were injected with either vehicle or with 16 mg/kg of Cisplatin (Fresenius Kabi Oncology, Homburg, Germany) intraperitoneally. All mice were injected intraperitoneally with 0.4% Trypan Blue (25 mg/kg) 2 hours prior to sacrifice. Thymuses were collected at 24 hours after Cisplatin administration and fixed in 4% paraformaldehyde (PFA) for 24 hours, sectioned at 70 µm with a Leica-VT 1000S Vibratome (Leica Biosystems, Nussloch, Germany), and further processed for immunofluorescent analysis.

### 2.5. Plasmids

Plasmids expressing the hyperactive Sleeping Beauty transposases *pCMV/SB11* and *sCMV/SB100x* were obtained from Addgene (#26552 and #34879, respectively. Watertown, MA, USA). DNA fragments encoding mouse myristoylated HA-tagged AKT and c-MYC proteins were obtained by PCR from the *pT3-EF1α-myr-HA-AKT* (#31789. Addgene, MA, USA) and *pcDNA3-cMYC* (#16011. Addgene, MA, USA) plasmids, respectively, and subcloned into the SfiI restriction sites of the Sleeping Beauty vector *pSBbi-puro* (#60523. Addgene, MA, USA). *pcDNA3 BAX* was obtained from Addgene (#8783. Addgene, MA, USA).

### 2.6. Hydrodynamic Tail Vein Injection (HDTVI)

For intrahepatic delivery of the transposon system, mice were hydrodynamically injected with 1 μg of *pCMV/SB11* or *pCMV/SB100x*, 4 μg of *pSBbi-puro-myr-AKT-HA*, and 10 μg of *pSBbi-puro-c-MYC* or 10 μg of *pcDNA3 BAX* via lateral tail vein. All plasmids were diluted in sterile-filtered 0.9% NaCl, and the total volume was adjusted to 10% (in mL) of the total body weight (in grams). All injected mice were monitored daily and sacrificed in groups at appropriate time points. All experimental and control groups contained 5 to 10 mice.

### 2.7. Trypan Blue Levels in Serum and Toxicity Analysis

A 0.4% Trypan Blue solution (T8154, Sigma-Aldrich, St. Louis, MO, USA was administered at different volumes (50, 100, 200, and 300 µL, corresponding to 6.5, 13, 25, and 40 mg/kg, respectively) via subcutaneous (SQ), intraperitoneal (IP), and intravenous (IV) injection to mice 2 hours prior to sacrifice or at the mentioned time. To quantify the concentration of TBlue in the blood, whole blood was withdrawn from humanely euthanized experimental animals by cardiac puncture into EDTA-coated tubes (Impromini EDTA, Medical Instruments, Milan, Italy) at 12, 24, 48, and 72 hours after the TBlue injection. Plasma was obtained through centrifugation at 4000 rpm for 20 minutes at 4 °C, and the samples were maintained at 2–8 °C while handling. A calibration curve was established using standard solutions prepared by diluting TBlue in mouse plasma, ranging from 10 to 400 ng/mL. The linear regression analysis of the data yielded a high correlation coefficient of 0.99, indicating a strong relationship between the TBlue concentrations and the corresponding measurements.

To determine whether circulating TBlue caused organ toxicity, mice received a single intraperitoneal injection of TBlue (26 mg/kg). Blood samples were collected by cardiac puncture into tubes without anticoagulant (Impromini Clot Activator, Medical Instruments, Milan, Italy) at 12, 24, 48, and 72 hours after the TBlue injection. The blood was centrifuged at 3500 rpm for 15 minutes at 4 °C to obtain serum. The serum was immediately transferred into clean 0.5 mL tubes. Urea, AST, ALT, ALP, and albumin levels were analyzed from 300 µL of serum using standard clinical colorimetric spectrophotometry.

### 2.8. Pulse and Chase

Trypan blue (T8154, Sigma-Aldrich, St. Louis, MO, USA, 0.4%) was injected at the indicated timepoint after injury. Then, a Green LIVE/DEAD Dye (L34970, Thermofisher, Waltham, MA, USA)was injected intravenously (100 μL/mouse) 2 hours before sacrifice.

### 2.9. Liver Sample Collection

Mice were sacrificed by carbon dioxide inhalation, and liver samples were collected at the described time points. Resected livers were fixed in 4% paraformaldehyde (PFA) for 48 hours, washed with phosphate buffered saline, and stored protected from light in 70% alcohol at room temperature.

### 2.10. Immunofluorescence

Livers, thymuses, and ears were embedded in 4% agarose and sectioned using the Leica-VT 1000S Vibratome (Leica Biosystems, Nussloch, Germany) at 100 µm for the liver and 70 µm for the thymus and ears. Tissue samples were blocked in 4% bovine serum albumin (BSA) in PBS for 2 hours at room temperature. Sections were incubated with primary antibodies overnight (dilution 1:100). The following day, sections were incubated with secondary antibodies conjugated with Alexa Fluor^®^ 488 or with Alexa Fluor^®^ 555 (Abcam, ab150073, ab150074. Cambridge, MA, USA, Dilution 1:500) for 2 hours at room temperature in the dark. Nuclei were counterstained with 4′,6-diamidino-2-phenylindole (DAPI) (D9542, Sigma-Aldrich, St. Louis, MO, USA). Sections were mounted and analyzed by confocal microscopy. The injury was detected non-specifically by overnight incubation with Donkey anti-mouse IgG (Alexa Fluor^®^ 488 ab150073 or Alexa Fluor^®^ 555 ab150074, Abcam, Cambridge, MA, USA). Dead areas were identified with Trypan Blue (TBlue) at 680 nm. To identify apoptotic dead cells, we used an anti-cleaved Caspase-3 (cCasp3) antibody (ab2302. Abcam, Cambridge, MA, USA).

### 2.11. iDISCO+ Clearing of Mouse Liver

Clearing of mouse liver lobes was performed as described earlier, with small adaptations [9]. Briefly, after immersion fixation in 4% paraformaldehyde, caudate lobes were stored in 70% ethanol pending analysis. After further dehydration in methanol/water series (80–100%), 1 hour each, the samples were cleared by incubation in 66% dichloromethane (270997. Sigma Merck KGaA, Darmstadt, Germany) and 33% methanol for 3 hours and twice in 100% dichloromethane for 30 minutes. The cleared liver lobes were then transferred to 100% benzyl ether (Sigma 108014) for imaging and long-term storage.

### 2.12. Light Sheet Fluorescence Microscopy

Cleared liver lobes were imaged with a light sheet microscope (Ultramicroscope II, Lavision Biotec, Bielefeld, Germany) and the ImSpector 347 software (Abberior, GöttingenGermany). All images were acquired using a 2× objective lens with an additional 0.8× zoom body, an NA of 0.6 (XLPLN10XSVMP, Olympus Corporation, Tokyo, Japan), and a sCMOS camera (Hamamatsu ORCA-Flash4.0, Shizuoka, Japan). Excitation laser lines of 647 nm were used to visualize Trypan Blue. The images were obtained with an xyz resolution of 3.80 μm.

### 2.13. 3D Image Reconstruction and Analysis

The serials of 16-bit uncompressed tif images were converted to an IMS file using the Imaris File Converter (Bitplane, Belfast, UK), and the 3D vision of acquisitions was reconstructed using the Imaris 9.7.2 software (Bitplane, Belfast, UK). The original light sheet scan was reduced using 3D crop, and measurements of the segmented volumes were obtained using semi-automatic surface creation on original images in Imaris as follows: A surface was created with a 7.55 μm grain size to cover the entire tissue (=total volume). Then, a second high-resolution surface was created based on the absolute intensity of TBlue to fully cover the TBlue immunosignal (7.55 μm grain size; = TBlue volume). Finally, the volume (μm^3^) of both surfaces was calculated by Imaris, and the percentage of TBlue+ volume in the tissue was determined. To reconstruct the main portal and central veins, IMS files with TBlue and 488 autofluorescence channels were cropped, and the veins were manually segmented with the ‘manual surface creation’ function based on the 488 tissue autofluorescence. For this, both the contour of the portal vein identified by the presence of bile ducts and arteries) and the central vein (identified by the absence of other hollow structures) were circled in each consecutive 10 μm for 350 μm. Snapshot images (1200 dpi) and animation movies of the cropped images were taken for visualization of TBlue and segmented surfaces.

## 3. Results

### 3.1. TBlue Marks Dead Cells during Liver Injury and Regeneration 

We induced liver injury and tested the ability of in vivo administration of Trypan Blue (TBlue) to label dead cells. For this, we administered a single intraperitoneal dose of carbon tetrachloride (CCl_4_; 1 mL/kg) in 30-g adult mice to cause centrilobular necrosis [10]. Subsequently, we administered 200 µL of 0.4% TBlue (25 mg/kg) via intravenous injection (IV) two hours before sacrificing mice at 12, 24, 48, 72, and 96 hours post CCl_4_ administration (Figure 1B–D and Appendix A). We detected its fluorescent peak emission of 660 nm in histological sections of fixed livers, which were co-stained with Alexa Fluor 488-labeled donkey anti-mouse immunoglobulins (IgG), which non-specifically label dead cells and endothelial cells (Figure 1A) [11,12]. At 12 hours after CCl_4_ administration, TBlue specifically marked a few scattered hepatocytes in the pericentral zone, which is the zone where CCl_4_ induces necrosis. The TBlue-positive hepatocytes were co-stained with IgG (Figure 1B). At 24 and 48 hours after CCl_4_ administration, TBlue marked the entire region of dead hepatocytes around the central vein (Figure 1C,D and Appendix A). Subsequently, the size of this TBlue positive region progressively decreased until it disappeared at 96 hours after CCl_4_ administration, a time when injuries regenerated and livers were largely normal (Appendix A). Notably, the administration of TBlue did not cause cell death in cells that were not affected by CCl_4_ or in livers not treated with CCl_4_ (Figure 1A–D). Moreover, unlike CCl_4_, administration of TBlue alone did not increase the levels of Aspartate Aminotransferase (AST), Alanine Aminotransferase (ALT), albumin, urea, or alkaline phosphatase in the serum, which shows that TBlue alone did not cause injury in the liver, kidney, or other organs even 72 hours after administration (Figure 1E–I). Thus, TBlue specifically marked dead cells in regenerating livers without causing liver toxicity at the used concentration. Moreover, the fact that TBlue labeled single hepatocytes in vivo highlights its high versatility and sensitivity. 

### 3.2. TBlue Detects Apoptotic Cell Death

It is thought that TBlue is not suitable for detecting apoptotic cells [13]. This is because TBlue primarily enters cells with compromised plasma membranes, and apoptotic cells can maintain membrane integrity during the early stages of apoptosis [13]. However, late apoptotic cells lose membrane integrity and become permeable to molecules like TBlue [13]. To determine whether TBlue can label apoptosis in vivo, we transfected plasmids expressing the pro-apoptotic gene BCL2 Associated X (BAX) and a reporter expressing green fluorescent protein (GFP) to label transfected hepatocytes by hydrodynamic tail vein injection (HDTVI) (Figure 1J–O) [14]. This strategy allowed us to genetically induce apoptosis in scattered hepatocytes [15] and to determine whether TBlue can label apoptotic cells. TBlue was administered and analyzed at 24 hours after HDTVI to label dead cells, and liver sections were co-stained with antibodies against GFP and cleaved Caspase 3 (cCasp3) (Figure 1K–O). TBlue labeled 95% of cells expressing *BAX* (Figure 1K), and, from those *BAX*-expressing hepatocytes, 98% were positive for cCasp3 and TBlue (Figure 1L,O), indicating that *BAX* expression triggered apoptotic cell death and that TBlue effectively detected apoptotic cells. In contrast, only 18% of mock-transfected hepatocytes expressing *GFP* but not *BAX* exhibited TBlue positivity (Figure 1K,M). In addition, we stained liver sections with anti-cCasp3 liver sections at 48 hours after CCl_4_ administration. As expected, most TBlue-positive dead hepatocytes were negative for cCasp3, consistent with CCl_4_ inducing death by necrosis [16]. However, a few scattered hepatocytes located at the periphery of the necrotic zone stained positive for cCasp3, indicating that these cells died by apoptosis (Appendix A). Importantly, these cCasp3-positive cells were also labeled by TBlue. Overall, these findings show that TBlue can label hepatocyte apoptosis in vivo.

### 3.3. In Vivo “Pulse and Chase” Tracking of Dead Cells

We used TBlue to monitor the dynamics of cell death and cell turnover during liver regeneration in pulse-chase experiments. To optimize the speed and sensitivity of cell death labeling, we compared different doses and routes of TBlue administration to mice with injured livers at 48 hours after CCl_4_ administration (Appendix A). Intraperitoneal (IP) injection of TBlue resulted in the strongest labeling of dead hepatocytes (Appendix A), followed by intravenous (IV) and subcutaneous (SQ) injection (Appendix A). Thus, intraperitoneal and intravenous administration were the most effective routes for TBlue delivery.

Next, we measured the kinetics of TBlue concentration in the blood plasma after injecting a single dose of TBlue. We calculated the half-life of circulating TBlue in plasma following intraperitoneal (IP) and intravenous (IV) injections (Figure 2A,B and Appendix A). We measured TBlue concentration by its absorbance at 590 nm in plasma samples collected at designated time intervals after mice were injected IP or IV with 200 µL of 0.4% TBlue (Appendix A). The peak plasma concentration was higher and occurred faster when TBlue was injected IV versus IP. Thus, after IV injection, the peak plasma concentration of TBlue was 160 ng/mL and occurred within 30 minutes, while after IP injection, the peak of 83 ng/mL occurred at 2 hours (Figure 2A,B). TBlue not only peaked faster but it was also cleared faster from circulation after IV injection (half-life of 2 hours) compared to IP (half-life of 18 hours). As a result of this short half-life, no hepatocytes were labeled when CCl_4_ was injected 24 or 48 hours after TBlue injection via IV (Appendix A). Thus, the residual TBlue present in mice 24 hours after TBlue injection by IV was not sufficient to label dead cells. In contrast, dead hepatocytes were labeled when CCl_4_ was injected 24 or 48 hours after TBlue was administered via IP (Appendix A). These experiments showed that IV administration of TBlue is suitable for acute pulse and chase in vivo labeling of dead cells, while IP injection has long-lasting perdurance and is useful for more continuous and cumulative labeling of dead cells.

We next used an IV injection of TBlue for “pulse and chase” experiments in vivo, where we labeled dead cells at a one-time point and tracked their persistence over time. We injected a single dose of TBlue at 24 hours after CCl_4_ administration and monitored labeled dead cells at 48 and 72 hours (Figure 2C,D). As a first approach to tracking the fate of labeled cells and to distinguish those cells that died at the time of TBlue administration from those that died later, we double-marked all dead cells by co-staining liver sections with IgG. In this way, TBlue-IgG double-labeling identified hepatocytes that died early but persisted in the tissue, whereas single IgG labeling identified hepatocytes that died after the time TBlue could label dead cells. Administration of TBlue via IV at 24 hours after CCl_4_ marked most IgG-positive dead cells detected at 48 hours after CCl_4_ but only a subset of all IgG-positive dead cells detected at 72 hours after CCl_4_, indicating that additional hepatocytes underwent necrosis after the time window were circulating TBlue could effectively label cell death (Figure 2D). Notably, dead hepatocytes labeled with TBlue at 24 hours after CCl_4_ administration remained detectable at 48 and 72 hours after CCl_4_, showing that clearance of dead hepatocytes is a slow and late regeneration event (Appendix A). In contrast, when TBlue was injected at 48 hours after CCl_4_ administration and cell death was analyzed at 72 hours after CCl_4_, all dead cells were double positive for TBlue and IgG, which shows that by 48 hours after CCl_4_, most of the cell death had already happened (Figure 2E,F). Importantly, TBlue administration did not prolong the time required for normal liver regeneration, as livers from TBlue and vehicle-injected mice were normal by 96 hours after CCl_4_ (Appendix A) [11]. Thus, TBlue-labeled cells can retain the dye for several days, showing that TBlue can acutely label dead cells in vivo and allow the tracking of labeled cells over extended periods of time. 

### 3.4. In Vivo “Pulse and Chase” of Two Populations of Dead Cells

We investigated the dynamics of dead cell clearance during liver regeneration in more detail. We marked and tracked dead hepatocytes at two different time points after CCl_4_ injection by first administering TBlue and, subsequently, a second dye. Furthermore, we utilized cholestatic livers because cholestasis slows down the clearance of dead hepatocytes due to impaired macrophage function [11]. This effect thus prolonged the persistence of necrotic corpses for over 72 hours after CCl_4_ administration and allowed the labeling and tracing of early versus late necrotic cells. We induced cholestasis by deleting *Yap* and *Taz* in bile duct cells, which disrupts the function and stability of the biliary tree [11]. Thus, we crossed *Opn-iCre^ERT2^* mice with *Yap^fl/fl^; Taz^fl/fl^* mice and administered tamoxifen to selectively delete *Yap* and *Taz* in the bile ducts of adult mice (Figure 2G,H). After a 3-week period, we injected CCl_4_ and labeled the first population of dead cells with TBlue via IV injection at 24 hours after CCl_4_ administration. The second population of dead cells was marked by injecting Green Dye, a Calcein AM and Ethidium homodimer-1-based vital dye, via IV two hours prior to the 7*2*-h endpoint.

*Yap/Taz* mutant livers exhibited extensive TBlue-labeled areas of injury at 72 hours after CCl_4_ administration, with many of the dead hepatocytes co-staining with Green Dye (Figure 2H). Double-labeled hepatocytes are those that died early but persisted in *Yap/Taz* mutant livers for at least 72 extra hours [11]. The presence of hepatocytes only labeled with Green Dye and not with TBlue revealed that additional cell death occurred after the 24-h period in which TBlue could effectively label dead hepatocytes. In contrast, non-cholestatic wild-type livers already regenerated at 96 hours after CCl_4_ administration, and no necrotic corpses positive for either dye were observed (Figure 2H). Only a few immune cells positive for TBlue but negative for Green Dye were detected (Figure 2H). Thus, subsequent intravenous injections of TBlue and Green Dye can mark and track two distinct subsets of dead cells. 

### 3.5. Three-Dimensional Mapping of Large Necrotic Areas in Whole Livers

Necrosis often manifests as extensive damage in entire regions of affected tissues [14]. Unfortunately, current labeling techniques lack the ability to precisely image and map these areas in whole organs. Hence, we assessed whether TBlue labeling could generate a three-dimensional (3D) map of liver injury following toxic hepatic injury. Therefore, to create a 3D map of liver injury, we cleared caudate lobes from CCl_4_-injured livers using iDisco+ (Figure 3A–F and Appendix A), a tissue clearing method compatible with immunolabelling and fluorescent microscopy [17]. The cleared liver lobes were then imaged using light sheet microscopy to obtain a volumetric map of the TBlue signal from injured livers at 48 hours after CCl_4_ liver injury (Figure 3B). Then, we rendered a three-dimensional surface model that depicted the geometry of the injury zones, their boundaries, and their spatial relationship with cells in the non-injured parts (Figure 3B). We added the surfaces of the segmented portal and central veins. This analysis showed that the regions of injured tissue marked by TBlue aligned with the central vein but not with the portal vein, which was supported by the lack of overlap of the biliary tree marked by the expression of mucin and the zone of pericentral injury (Figure 3D–G). Measuring the volume of the injured regions based on their surfaces determined that, at the peak of injury, the total volume of dead tissue marked by TBlue accounted for approximately 33% of the liver’s total volume (Figure 3E,F). Hence, TBlue staining effectively mapped extensive areas of cell death within entire injured livers. 

### 3.6. Detection of Cell Death in Liver Cancer Using TBlue Labelling

Understanding the dynamics of cancer cell death during tumor progression or regression is critical for the assessment of therapy efficacy. Thus, we determined whether TBlue could detect dead cancer cells in liver tumors. To test this, we induced liver cancer by transfecting Sleeping Beauty constructs encoding for the oncogenes c-MYC and AKT (constitutive active by myristoylation and tagged with HA) into normal hepatocytes (Figure 4A). This transfection strategy targeted approximately 2–5% of hepatocytes and led to the formation of multiple tumors within 6 weeks (Figure 4B,C) [18]. *c-MYC;AKT*-expressing hepatocytes formed cholangiocarcinoma (CCA), as revealed by the glandular morphology of the tumors and by the expression of the cholangiocyte marker Osteopontin (OPN) (Figure 4B–D). 

To detect cell death in mature tumors, we administered 200 µL of 0.4% Trypan Blue via IP injection 2 hours before sacrifice at the sixth week of tumor development (Figure 4A). While whole livers with tumors from mice that were not injected with TBlue exhibited a yellowish appearance (Figure 4B), livers injected with TBlue displayed a distinct blue coloration localized to the regions where CCA tumors were present (Figure 4C). Anti-HA staining to label HA-tagged AKT-positive cells showed that TBlue specifically labeled dead cancer cells, which were mainly in the lumen of CCA tumors in vehicle-injected mice (Figure 4E). Furthermore, TBlue-labeled tumor cells were also positive for IgG (Figure 4G), supporting the fact that TBlue labels dead cancer cells. Altogether, these findings demonstrate the utility of TBlue labeling as a tool for detecting cell death in liver cancer.

Next, we determined whether TBlue could detect chemotherapy-triggered cancer cell death in liver tumors. We administered a single dose of Cisplatin (10 mg/kg) to tumor-bearing mice to trigger the death of proliferating cancer cells at 5 weeks of tumor development. We injected 200 µL of 0.4% TBlue via IP injection 2 hours before sacrifice at 6 weeks of tumor development. Anti-HA tag staining, which labeled AKT-positive cells, showed that TBlue specifically marked dead cancer cells, which localized throughout the tumor in cisplatin-treated mice. In contrast, dead cancer cells in vehicle-treated mice were primarily localized at the lumen of CCA tumors (Figure 4E–H). Altogether, these findings demonstrate the utility of TBlue as a valuable tool for the in vivo understanding of the dynamics of cancer cell death during tumor progression and therapy-induced regression, both critical aspects for the assessment of therapy efficacy in animal models.

### 3.7. Detection of Cell Death in Other Organs Using TBlue Labeling

We next tested the effectiveness of TBlue in labeling acute cell death in other mouse models of organ injury, such as tissue damage triggered by ischemia-reperfusion, chemotherapy-induced toxicity, and organ atrophy caused by fasting. First, we induced injury by ischemia-reperfusion, which occurs when transient obstruction of blood flow (ischemia) is followed by a subsequent re-oxygenation of the tissues (reperfusion). We used a non-invasive magnet clamping method [8] to cause ischemia-reperfusion on the mouse ear skin to determine if TBlue can mark dead cells in vivo (Appendix A). TBlue marked dead cells in several cell layers of the ear, including the cartilage and dermis (Appendix A). Thus, TBlue effectively marked dead cells after ischemia-reperfusion. 

Cisplatin and other chemotherapy-cytotoxic drugs can cause thymic atrophy by directly killing rapidly dividing thymocytes (immature T cells) [19,20]. Thus, we evaluated whether TBlue can label single thymocytes eliminated by a high and acute dose of cisplatin (16 mg/kg) (Appendix A). We found that a single dose of cisplatin was sufficient to cause thymocyte cell death and that TBlue effectively labeled dead thymocytes (Appendix A). Thus, similar to the effect of cisplatin, prolonged fasting can reduce the total count of thymocytes [21], so we analyzed the thymus of mice at 48 hours of fasting and found that TBlue labeled many dead cells in the thymus. In contrast, thymocytes from mice injected with vehicle or that fed ad libitum did not present any TBlue labeling, demonstrating that TBlue efficiently detects dead thymocytes without causing overt toxicity effects. Altogether, this data showed that TBlue labeling readily detects dead cells in different organs killed by different insults, thus highlighting the versatility of using TBlue as an effective method to mark cell death in vivo. 

## 4. Discussion

Current cell death labeling methods, such as TUNEL, Annexin V, and caspase activity assays, are valuable for detecting cell death in vitro [22]. However, most of these methods are not suitable for labeling and tracking dead cells in vivo. Trypan Blue (TBlue) staining has traditionally been used in settings that use cultured cells or cells from dissociated tissues, but here we demonstrated that TBlue injection is a highly versatile and specific method for labeling dead cells in vivo. We showed that TBlue allows labeling and tracking dead cells in injured and cancerous tissues with single-cell resolution in a whole organ. 

How do the advantages and disadvantages of TBlue labeling compare with those of other cell death labeling methods? First, Trypan Blue may be used to detect different types of cell death, including necrosis, apoptosis, and likely other types of cell death that perturb membrane integrity. In contrast, most current cell death labeling methods, like Annexin V and caspase activity assays, have limitations in detecting different forms of cell death [22]. While caspase activity is specific to apoptosis [23,24], Annexin V can detect apoptosis and anoikis [25,26]. However, neither method can detect other types of cell death, such as necrosis [13], ferroptosis [27], or pyroptosis [28], that also compromise the integrity of the cell membrane and are predicted to be detected by TBlue. Thus, Trypan Blue has the potential to detect a broad spectrum of cell death types when membrane integrity is compromised. 

Second, TBlue can be used to detect cells that no longer have DNA. In contrast, other commonly used cell death labeling methods, such as Hoechst, DAPI, and Propidium Iodide, and enzymatic assays that detect DNA double-strand breaks, such as TUNEL (Terminal deoxynucleotidyl transferase dUTP nick-end labeling), rely on the use of fluorescent DNA-binding dyes [29]. These methods are limited by the presence of DNA fragments for the dyes to bind and emit a signal [29]. However, DNA degradation can occur many hours and even days before cell corpses are cleared from the injured tissue. Therefore, while these methods cannot detect cellular corpses that have lost their genomic DNA, such as necrotic hepatocytes in CCl_4_-treated livers, TBlue can effectively label dead hepatocytes for prolonged periods (Figure 1A and Appendix A) [11,30]. In addition, these methods may not be specific to cell death, as DNA fragmentation can also occur during DNA repair and replication [29]. Therefore, even though these methods are general enough to detect different types of cell death, their utility is limited to cells that still have some integrity in their genomic DNA. In contrast, Trypan Blue can readily label necrotic corpses even days after the genomic DNA has been degraded. 

Third, and perhaps the biggest advantage, is the ability of TBlue to perform in vivo detection and tracing of dead cells in pulse and chase experiments. This is thanks to its high sensitivity and the short duration of its circulation, thus enabling researchers to follow the fate of dead cells in vivo and to understand the dynamics of cell death and clearance over specific time intervals. Importantly, the fact that TBlue can be effectively combined with other viability dyes, such as Green Dye, offers a robust method for distinguishing distinct dead cell populations and their evolution over time. This capability makes Trypan Blue a powerful and flexible tool for investigations into the intricate processes of homeostasis, regeneration, and cancer, providing valuable insights into the fate of cells in vivo.

Notably, although previous studies reported that high doses of TBlue can be toxic in vitro and in vivo in various organisms [28,31,32,33,34], our findings indicate that the doses of TBlue administered in our study did not cause overt phenotypes. In adult mice, an LD50 of 267 mg/kg was reported for subcutaneous injection of a 1% trypan blue solution [28]. In contrast, we mainly used TBlue at a dose of 25 mg/kg, which is 10 times lower than the reported LD50 for mice and did not cause any adverse phenotype in the liver, kidney, or other organs. Our findings thus emphasize the importance of considering low dosage levels to avoid the potentially toxic effects of TBlue and highlight its value in the in vivo labeling of dead cells.

In conclusion, while TBlue staining is commonly used for in vitro applications to detect cell death, we have demonstrated its potential for labeling, tracking, and mapping cell death in vivo. TBlue can aid in understanding the progression of diseases or evaluating the efficacy of novel therapeutic approaches in diverse animal models, providing valuable insights into the homeostasis and regeneration of different tissues.

## Figures and Tables

**Figure 1 cells-13-01379-f001:**
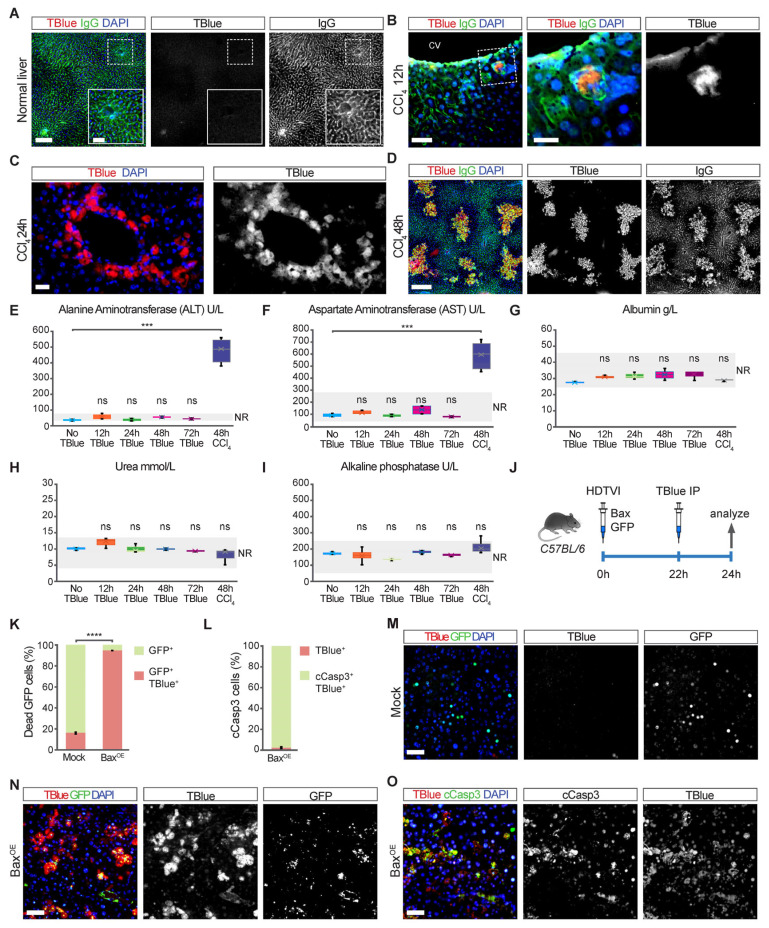
**TBlue labels necrotic and apoptotic cells in vivo.** (**A**) The liver section of a normal liver was injected with TBlue and stained for anti-mouse-IgG (green) showing normal sinusoids. Scale bars, 250 μm. Insert showing a magnified view of the normal liver tissue and the absence of cell death. Scale bars, 100 μm. (**B**–**D**) Liver sections of injured livers at 12, 24, and 48 hours after CCl_4_ administration from mice injected with TBlue (red) to mark dead hepatocytes. Staining with anti-mouse IgG (green) showing normal sinusoids and areas of injury. Scale bars, 100 and 40 μm for (**B**), 100 μm for (**C**), and 250 μm for (**D**). (**E**–**I**) Serum enzyme analysis showing the levels of ALT, AST, Alb, Urea, and ALP. NR, normal range (**J**) Experimental outline: C57BL/6 mice were hydrodynamically injected with the *BAX* and *H2B-GFP* plasmids. After 24 hours, all mice were euthanized and their livers processed for further staining. (**K**,**L**) Quantification of dead H2B-GFP possitive cells marked with TBlue and TBlue-labelled hepatocytes positive for cleaved caspase-3 after in vivo transfection of *BAX* by hydrodynamic tail vein injection. (**M**–**O**) Immunofluorescent staining of liver sections containing dead hepatocytes marked with TBlue and co-stained with cleaved caspase-3 after in vivo transfection of *BAX* by hydrodynamic tail vein injection. TBlue (dead hepatocytes, red), H2B-GFP (transfected hepatocytes, green), and DAPI (nuclei, blue) (scale bars, 100 μm). Data are means ± SEM. *** *p* < 0.001, **** *p* < 0.0001; n.s., not significant.

**Figure 2 cells-13-01379-f002:**
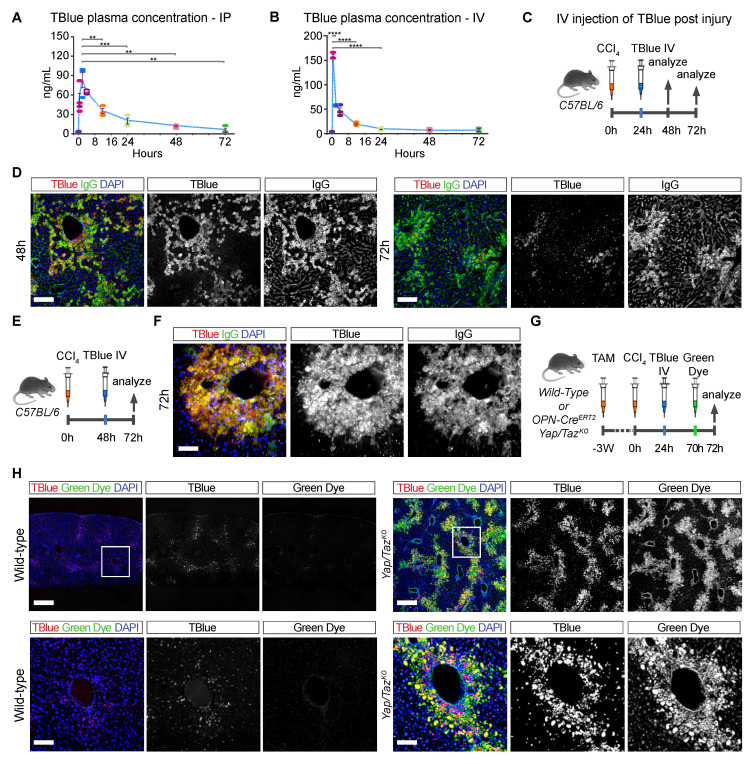
**TBlue is suitable for pulse and chase experiments in vivo.** (**A**,**B**) Time course showing the concentration of circulating TBlue in plasma from mice after intraperitoneal and intravenous administration. Plasma was collected at 0.5 (purple), 1 (pink), 2 (blue), 4(crimson), 12 (orange), 24 (light green), 48 (light pink) and 72 (green) hours after TBlue administration. (**C**) Experimental outline: C57BL/6 mice injected with TBlue at 24 hours post-CCl_4_. All mice were euthanized, and their livers were processed for further staining at the indicated times. (**D**) Immunofluorescent staining of liver sections showing dead hepatocytes marked with TBlue (red), IgG (green), and DAPI (blue). Scale bars, 100 μm. (**E**,**F**) Experimental outline and immunofluorescent staining of liver sections from, C57BL/6 mice injected with TBlue at 48 hours post CCl_4_. TBlue (red), IgG (green), and DAPI (blue). Scale bars, 100 μm. (**G**,**H**) Experimental outline and immunofluorescent staining of liver sections from wild-type and *OPN-Cre^ERT2^; Yap^fl/fl^;Taz^fl/fl^* mice injected with tamoxifen to induce the recombination of floxed alleles and then injected with CCl_4_ to cause injury. TBlue and Green Dye were injected at different time points post CCl_4_ to mark different populations of dead hepatocytes. TBlue (red), IgG (green), and DAPI (blue). Scale bars, 100 μm. Data are means ± SEM. ** *p* < 0.01, *** *p* < 0.001, **** *p* < 0.0001.

**Figure 3 cells-13-01379-f003:**
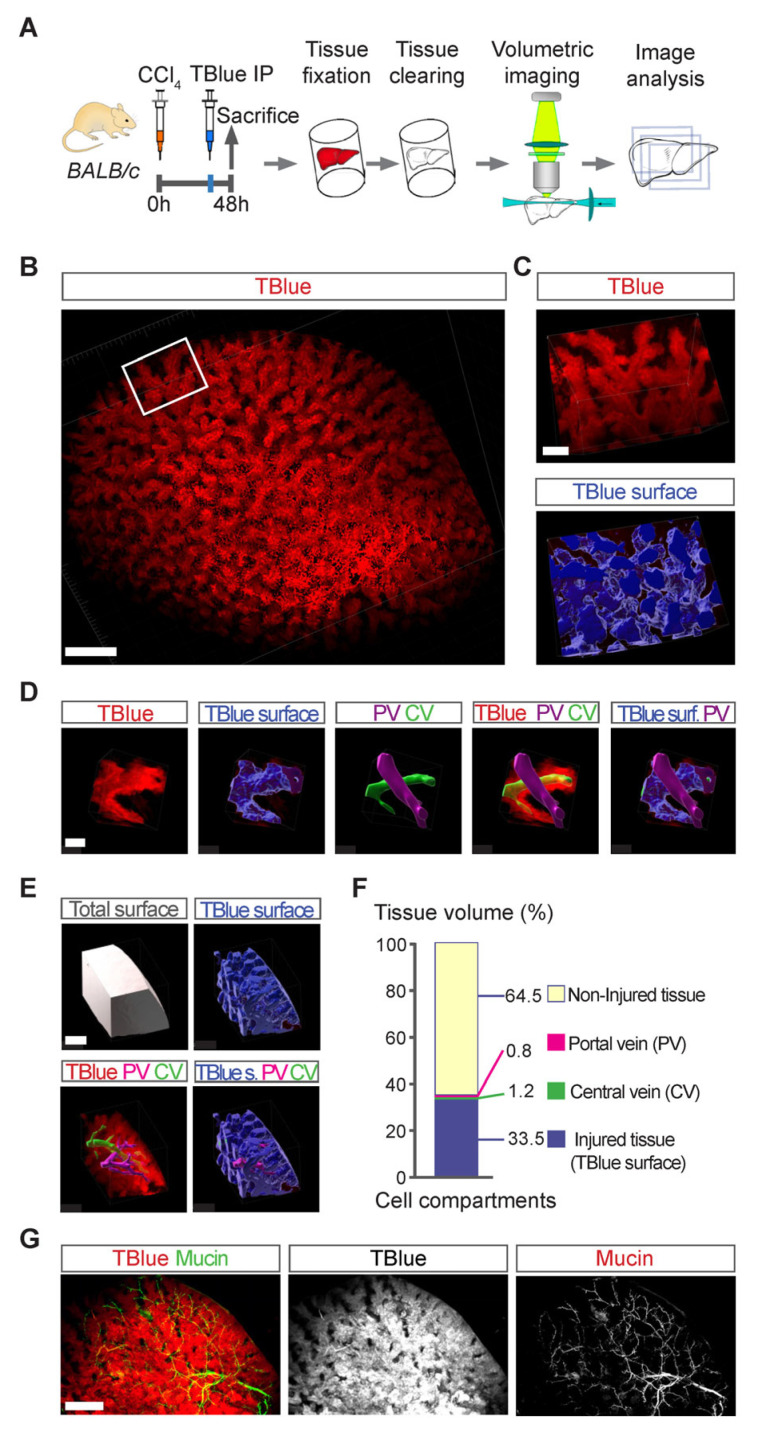
**Tridimensional (3D) mapping of liver injury using TBlue.** (**A**) Experimental outline: C57BL/6 mice were injected with TBlue at 2 hours prior to sacrifice at 48 hours post CCl_4_. All mice were euthanized, and their livers were processed for organ clearing with iDisco and light sheet microscopy. (**B**) 3D visualization of the injury areas of iDisco cleared livers imaged by light sheet microscopy. TBlue (red) shows a 3D reconstruction of injured tissue in the whole caudate lobe cleared with iDisco at 48 hours after CCl_4_. Scale bar, 1000 μm. (**C**) Higher magnification of a cropped region of the 3D image in (**B**). Scale bar, 500 μm. (**D**) High magnification area showing the surface created based on TBlue (blue), and the surfaces of the segmented Portal vein (PV; magenta) and Central vein (CV; green). Scale bar, 200 μm. (**E**) Region of the 3D image showing surface volumes of the total tissue surface (white) and TBlue surface (blue). (**F**) Quantification (Imaris) of the volume from the TBlue surface over the total surface from the image shown in (**E**). Scale bar, 200 μm. (**G**) 3D visualization of the biliary tree (Mucin, green) and the injury area in iDisco cleared livers imaged by light sheet microscopy. TBlue (red) shows a 3D reconstruction of injured tissue in the caudate lobe cleared with iDisco at 48 hours after CCl_4_. Scale bar, 1000 μm.

**Figure 4 cells-13-01379-f004:**
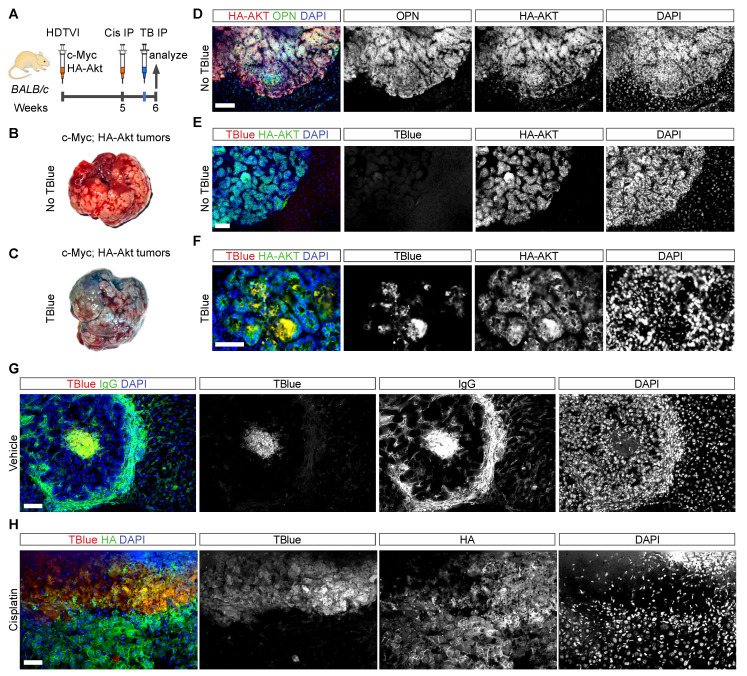
**Detection of liver cancer cell death upon Cisplatin treatment.** (**A**) Experimental outline: BALB/c mice injected with TBlue at 6 weeks of tumor development induced by HDTVI of Sleeping Beauty plasmids encoding for the oncogenic c-MYC and AKT proteins. All mice were euthanized and their livers were processed for further staining at the indicated times. (**B**,**C**) Whole mount livers with cholangiocarcinoma from mice injected with saline or TBlue. (**D**) Immunofluorescent staining of liver sections showing the expression of the cholangiocarcinoma marker Osteopontin (OPN, green), HA-AKT (red), and DAPI (blue). Scale bar, 250 μm. (**E**,**F**) Immunofluorescent staining showing TBlue-positive dead cholangiocarcinoma cells as marked by HA-AKT (green), TBlue (red), and DAPI (blue). (**E**) Contains TBlue (red) and IgG (green) showing dead cells in the core of cholangiocarcinoma tumors. Scale bar, 250 μm. (**F**) Immunofluorescent staining of liver sections showing massive cell death in cholangiocarcinoma tumors from mice treated with one dose of Cisplatin. HA-AKT (green), TBlue (red) and DAPI (blue). Scale bar, 250 μm (**G**,**H**).

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
