# Peer review of "In Vivo Tracking and 3D Mapping of Cell Death in Regeneration and Cancer Using Trypan Blue"

_cells, 2024, doi:10.3390/cells13161379_

Round 1

Reviewer 1 Report

Comments and Suggestions for Authors

The manuscript reports a procedure to evaluate cell death in vivo in the rat liver subjected to distinct treatments, including cancer. The study includes an extensive methodology that supports the usefulness of TP for the study of cell death in the liver. However, the basic data on the use of TP to detect cell death are widely known in the literature and the usefulness of the information from this study could be restricted to the liver degeneration model. For example, as the authors mention, the toxic effect of TP has been reported in other tissues, and it is not clear if the methodology and doses used in the study would be applicable to other models.

Regarding the specificity of TP in marking apoptosis, it is possible that the observations shown illustrate necrosis secondary to apoptotic processes. A complementary aspect that could be made is the use the technique in embryonic models in which cell death has been characterized as apoptotic (for example the interdigits of the developing limb). I would suggest to the authors a similar approach using their technique in avian embryos that is an quick and easy assay, to confirm the usefulness of the technique in other in vivo models.

Otherwise, my suggestion is to modify the title of the study to mention that the objective of the study is the development of an in vivo technique to monitor cell death in the liver using TP.

Whatever the authors decide about the manuscript I strongly suggest that the tittle should contain the term “Trypan Blue”

Control in Fig. 1 (A) should be shown at the same magnification than experimental samples (Fig 1B, Fig. 2…)

Author Response

Reviewer 1:

The manuscript reports a procedure to evaluate cell death in vivo in the rat liver subjected to distinct treatments, including cancer. The study includes an extensive methodology that supports the usefulness of TP for the study of cell death in the liver.

We thank the reviewer for his/her enthusiasm.

However, the basic data on the use of TP to detect cell death are widely known in the literature and the usefulness of the information from this study could be restricted to the liver degeneration model. For example, as the authors mention, the toxic effect of TP has been reported in other tissues, and it is not clear if the methodology and doses used in the study would be applicable to other models.

While it is true that TBlue has been extensively used and reported in vitro and ex vivo, there are no in vivo reports of the use of TBlue to mark single cells in pulse and chase experiments or to do 3D mapping of whole injured organs.

We used the mouse liver due to its clinical relevance and because of the great versatility it offers regarding experimental tools to study cell death. However, we are convinced that our findings in the liver can be easily extrapolated to other organs and other types of injury. Now, we show that the usefulness of in vivo labelling of dead cells with TBlue is not limited to the liver but can be used to label dead cells in other organs (such as thymus and skin) and triggered by other insults (such as ischemia-reperfusion, chemotherapy-induced toxicity and organ atrophy, as shown in supplementary figures 6 & 7). Thus, we are confident that the impact and usefulness of this new method will go well beyond the liver.

In addition, we have now included additional toxicity assays that measure the presence of injury markers in the serum at different timepoints after TBlue administration. While we did not find any evidence of overt phenotypes caused by TBlue toxicity, it is possible that other organs and markers we did not analyze might be slightly affected. However, as we mention in the discussion, the doses we employ are 10 times lower than the reported toxic doses in vivo, so we are confident that the in vivo labelling of dead cells using TBlue can be largely safe and accurate.

Regarding the specificity of TP in marking apoptosis, it is possible that the observations shown illustrate necrosis secondary to apoptotic processes. A complementary aspect that could be made is the use the technique in embryonic models in which cell death has been characterized as apoptotic (for example the interdigits of the developing limb). I would suggest to the authors a similar approach using their technique in avian embryos that is a quick and easy assay, to confirm the usefulness of the technique in other in vivo models. Otherwise, my suggestion is to modify the title of the study to mention that the objective of the study is the development of an in vivo technique to monitor cell death in the liver using TP.

We agree with the reviewer that apoptosis can lead to cell-autonomous secondary necrosis in unusual circumstances, such as when apoptotic proteins are present in low levels or when apoptosis is experimentally inhibited. However, in our assays we overexpressed high levels of BAX, a pro-apoptotic protein that causes cytochrome C release due to mitochondrial permeabilization, and induction of the intrinsic apoptotic pathway through Caspase 9 and 3 activation. The likelihood of a BAX-overexpressing cell to not completing apoptosis and instead proceeding to secondary necrosis is extremely low, as already injecting saline solution by hydrodynamic tail vein injection is sufficient to trigger hepatocyte apoptosis1.

Nonetheless, using avian embryos to mark interdigit apoptosis could indeed support that TBlue administration marks apoptotic cells. However, we find it technically challenging for our application as interdigit apoptosis in chick embryos occurs when the embryo barely measures 9 to 18 mm in length (between embryonic days 5 and 7). Injection into the chorioallantoic vein is a possible route of TBlue administration, yet, to our knowledge, it is usually performed at embryonic days 10 to 12, when the size of the embryos is considerably larger and the chorioallantoic membrane is fully developed. Unfortunately, interdigit apoptosis only can be evaluated during embryonic days 5-7, so testing the usefulness of TBlue in this setting, not only goes beyond the scope of this manuscript, but will also require the development of new methodological approaches.

Excitingly, we have now shown that TBlue can label dead cells in the thymus and ear skin upon ischemia-reperfusion injury, chemotherapy-induced toxicity and organ atrophy caused by fasting. Therefore, we are confident that labelling dead cells in vivo with TBlue will prove to be a very versatile method that will further our understanding of physiological and pathological cell death types in many different organs.

Whatever the authors decide about the manuscript I strongly suggest that the tittle should contain the term “Trypan Blue”

Following the reviewer´s suggestion, we now changed the tittle to ”In Vivo Tracking and 3D Mapping of Cell Death in Regeneration and Cancer using Trypan Blue”

Control in Fig. 1 (A) should be shown at the same magnification as experimental samples (Fig 1B, Fig. 2…)

The aim of showing the low magnification was to demonstrate that TBlue alone was not sufficient to cause cell death across different liver zones. Nevertheless, we have now followed the reviewer´s advise and added an inset magnifying a central vein region of the very same control picture, which shows the absence of TBlue labelling in greater detail and with a similar magnification to Fig 1B. 

Reviewer 2 Report

Comments and Suggestions for Authors

I think the authors have made a strong case for their work. It would be useful to extend the examples provided. Is the approach good for detecting cell death in the mouse thymus? Does the dye enter the brain? Does if show patterns of cell death duing inflmmation (.g. lung/brain/bwel,,,)? How early can TBlue detect cardiac infarction or a renal perfusion defect from a clamped artery etc. 

While the extra wrok may delay publication of the current fidndings, I think the eextra results will add to the numbe of citations the paper receives.

Minor point

This sentence  may not be communicating what the authors intended 

"Anti-HA tag staining labelled AKT-expressing cells showed that TBlue specifically eliminated cancer cells." as it implies TBlue is capable of kiiling cells.

Author Response

Reviewer 2:

I think the authors have made a strong case for their work. It would be useful to extend the examples provided.

We thank the reviewer for his/her enthusiasm and positive comments. We now show the utility of TBlue labelling in other injury models such as ischemia-reperfusion, organ atrophy and chemotherapy induced toxicity.

Is the approach good for detecting cell death in the mouse thymus?

Thanks to the reviewer´s suggestion, we now present an additional figure showing TBlue labelling of dead cells in the thymus by two types of insults: chemotherapy-induced thymus injury and fasting-induced thymus atrophy. We present these data in supplementary figure 7. These results certainly expand the utility and impact of this methodology.

Does the dye enter the brain?

Interestingly, the discovery of the blood brain barrier, by Edwin Goldmann, was made possible by injections of TBlue either in the bloodstream or in the cerebrospinal fluid2. Goldmann found that TBlue did not penetrate into the brain when injected in the blood and did not diffuse into the body when injected in the cerebrospinal fluid. Therefore, systemic administration of TBlue cannot be used to detect dead cells in the brain, nor will cause toxicity in the neural tissue.

Does if show patterns of cell death during inflammation (.g. lung/brain/bwel,,,)? How early can TBlue detect cardiac infarction or a renal perfusion defect from a clamped artery etc. 

Thanks to the reviewer´s suggestion, we addressed this question by performing a non-invasive ischemia-reperfusion (IR) assay in the skin of the ear to determine if TBlue can label early cell death triggered by IR. We present these data in supplementary figure 6, where we show that TBlue effectively labels dead cells in the cartilage and dermis layers of the ear, even only after 2 hours post reperfusion. These results showed that TBlue can detect early cell death events after acute injury in the skin. Moreover, and, together with the thymus data, these results greatly expand the impact and use of this methodology.

While the extra work may delay publication of the current findings, I think the extra results will add to the number of citations the paper receives.

We certainly agree with the reviewer and thank him/her for the insightful comments and suggestions.

Minor point

This sentence may not be communicating what the authors intended 

"Anti-HA tag staining labelled AKT-expressing cells showed that TBlue specifically eliminated cancer cells." as it implies TBlue is capable of killing cells.

Thanks for pointing this mistake out. We have now corrected the sentence and replaced the word “eliminated” by “marked”, as the statement aimed at highlighting the specificity of TBlue in marking dead cancer cells, not at suggesting that TBlue can kill cancer cells.

References

1                Seehawer, M. et al. Necroptosis microenvironment directs lineage commitment in liver cancer. Nature 562, 69-75 (2018). https://doi.org:10.1038/s41586-018-0519-y

2               Bentivoglio, M. & Kristensson, K. Tryps and trips: cell trafficking across the 100-year-old blood-brain barrier. Trends Neurosci 37, 325-333 (2014). https://doi.org:10.1016/j.tins.2014.03.007